# The Role of Hydrochar in Promoting Methane Production from Anaerobic Digestion with Different Inocula

Jieyi Sun [1], Shicheng Zhang [1,2,3,4] and Gang Luo [1,2,3,4,*]

1   Department of Environment Science and Engineering, Fudan University, Shanghai 200433, China; 20210740022@fudan.edu.cn (J.S.); zhangsc@fudan.edu.cn (S.Z.)
2   Shanghai Key Laboratory of Atmospheric Particle Pollution and Prevention (LAP3), Department of Environmental Science and Engineering, Fudan University, Shanghai 200433, China
3   Shanghai Technical Service Platform for Pollution Control and Resource Utilization of Organic Wastes, Shanghai 200438, China
4   Shanghai Institute of Pollution Control and Ecological Security, Shanghai 200092, China
*   Correspondence: gangl@fudan.edu.cn

**Abstract:** Carbon materials, including hydrochar, can promote methane production from anaerobic digestion, and many previous studies have investigated the mechanism by analyzing changes in the microbial community. Based on the fact that the major functional microorganisms have varied in different studies, in order to investigate the effect of inocula on the final microbial composition and to compare the promotion effects of hydrochar on different microorganisms, this study introduced inocula from three distinct sources to anaerobic sequential batch reactors and analyzed the microbial community using 16S rRNA gene sequencing. Hydrochar showed significant promotion effects for all three of the inocula, by increasing microbial activity at high acid concentrations and/or by reducing butyrate accumulation. The dominant microorganisms in all reactors were *Clostridium* and *Methanosarcina*, and hydrochar increased the proportion of acetoclastic methanogens. The bacteria which were promoted by hydrochar (e.g., *Aminicenantales*, *Anaerolineaceae* and *SBR1031* in anaerobic granular sludge and sludge digestate, as well as *Lachnospiraceae* in food waste digestate) only accounted for around 10%. The bacteria found to be involved in DIET in earlier studies were lacking in this study. It was speculated that hydrochar acted as an electron intermediate and supported microbial aggregation, while the possibility that hydrochar promoted DIET cannot be ruled out.

**Keywords:** anaerobic digestion; hydrochar; microbial analysis

## 1. Introduction

Anaerobic digestion is a widely applied method to treat organic wastes in a green and clean way. Under anaerobic conditions, microorganisms convert organic wastes into biogas, which mainly comprises methane and can be utilized as fuel. During this process, the quantity of waste is reduced, and the energy of organic matter is recovered at the same time.

Although anaerobic digestion has been widely used, it has some problems, such as a slow reaction rate and the accumulation of organic acids, which may cause reactor failure. Some studies have found that the performance of the anaerobic reactor can be improved by adding carbon materials such as biochar and hydrochar [1], prepared by pyrolysis or the hydrothermal treatment of biomass waste. For example, in the sequential batch experiment, the addition of hydrochar can shorten the lag period of methane production and increase the methane production rate [2–4]. The addition of biochar to the continuous flow reactor can improve the methane yield [5,6], increase the organic load [7], and enhance the tolerance to shock loads [8].

There have been a large number of studies that tried to ascertain the mechanism by which biochar promotes anaerobic digestion. The theory that biochar promotes direct inter-

species electron transfer (DIET) is popular. DIET is conducted by e-pili and cytochrome C, which can directly transfer electrons between contacted bacteria and methanogens [9]. Compared to mediated interspecies electron transfer (MIET), which relies on electron shuttles such as hydrogen and formate, DIET is more efficient [10,11], and therefore increases the rate of methanogenesis, resulting in the improved efficiency and stability of anaerobic digestion. Biochar in the anaerobic reactor can help establish DIET by either acting as a conductor between cells, or by providing and receiving electrons with the functional groups on its surface [10]. A pure culture experiment is necessary for obtaining strict proof of the occurrence of DIET. Up until now, pure-strain co-culture experiments have revealed the involvement of *Geobacter* spp., *Syntrophus aciditrophicus*, and *Methanosaeta harundinacea* in DIET [12–14]. However, anaerobic digestion is usually a mixed-culture system and therefore, in many studies, the occurrence of DIET was inferred from the changes in the microbial community. For example, Zhao et al. and Wang et al. found that adding biochar enriched *Geobacter* [5,15]. Shao et al. found that there were more *Thermovirga* attached to biochar in their study using acetic acid as a substrate [16]. Li's experiment on the anaerobic digestion of cardboard found that the proportion of *Syntrophomonas* increased with the increasing addition of biochar [17]. The microorganisms enriched with carbon materials may be affected by many factors such as the substrate, the inoculum, and the properties of the carbon materials. However, there have been few systematic studies on the contribution of these factors to differences in the microbial composition of reactors.

Our previous study used hydrochar to promote methanogenesis with glucose as a substrate, and enriched *Trichoccocus* that may participate in DIET [2], while subsequent experiments under similar conditions enriched *Azospira* but lacked *Trichoccocus*, which may be due to the difference in the inocula [18]. The discrepancy between the two studies raises the question: Do differences in inocula affect the effectiveness of hydrochar in promoting methanogenesis? Does the pathway by which hydrochar promotes methanogenesis differ when different microorganisms are enriched? The differences in the responses of different inocula to hydrochar may reveal the key factors of hydrochar that alter microbiota composition and promote anaerobic digestion, which is worthy of investigation.

In order to further understand whether microorganisms enriched by hydrochar are associated with the inoculated sludge, inocula from three different sources was used to enrich anaerobic digester bacteria through sequential batch cultivation in this study. The effects of hydrochar on methane production and the differences in enriched bacteria were also investigated.

## 2. Materials and Methods

### 2.1. Hydrochar and Inoculum

Dehydrated sludge obtained from the secondary sedimentation tank of a wastewater treatment plant in Shanghai, China was used to prepare hydrochar. The process was described in detail in the previous paper [2]. In brief, the sludge was carbonized through hydrothermal treatment at 300 °C for 1 h and the derived hydrochar was washed using tetrahydrofuran, ethanol, and water, successively. The characteristics of the hydrochar are described in the supplementary material.

Three inocula were used in this study, namely anaerobic granular sludge (denoted as G), sludge digestate (denoted as S), and food waste digestate (denoted as F). Anaerobic granular sludge was collected from a UASB reactor that treated citric acid wastewater in Anhui, China, and crushed by a grinder before inoculation. Sludge digestate was collected from the anaerobic digester of a wastewater treatment plant in Shanghai. Food waste digestate was obtained from the anaerobic digestion reactor of a municipal waste treatment plant in Shanghai. The characteristics of the inocula are listed in Table 1. Preincubation at 37 °C for 10 days was performed to consume residual substrates before inoculation.

**Table 1.** The characteristics of the inocula.

| Inoculum | pH | TS (g/L) | VS (g/L) |
|---|---|---|---|
| G | $8.12 \pm 0.05$ | $50.4 \pm 6.8$ | $37.4 \pm 3.5$ |
| S | $8.39 \pm 0.05$ | $20.3 \pm 1.1$ | $17.2 \pm 0.9$ |
| F | $7.94 \pm 0.05$ | $34.5 \pm 0.7$ | $17.4 \pm 0.5$ |

*2.2. Reactor Setup*

The experiment was performed in sequential batch mode. Glass serum bottles were used as reactors with the volume of 118 mL, each containing 50 mL liquid. The amount of inoculum was 4 g VS/L in all cases. The composition of the substrate was as follows: 5 g/L glucose, 2 g/L sodium bicarbonate, and BA Medium containing nutrients and trace elements, prepared according to the previous study [19]. Each reactor was flushed with nitrogen for 5 min and then sealed with a nitrile rubber stopper and incubated at 37 °C. For each inoculum, the experimental groups with 1 g/L hydrochar (denoted as G1, S1, F1), and the control groups without hydrochar (denoted as G0, S0, F0), were set up. All the experiments were performed in triplicates.

In total 4 batches were conducted in sequence with a total duration of 86 days in order to fully enrich the functional microorganisms. When a batch was finished, the liquid was removed by centrifugation to retain the solid which contained microorganisms and hydrochar, and the new substrate was added to start the next batch. A figure explaining the reactor configuration is attached in the supplementary material (Figure S1). During the last batch, the liquid and microorganisms in the reactor were extracted using a syringe every 2–3 days and, after centrifugation, the supernatant was stored at −20 °C until analysis, and the precipitated part was stored at −80 °C.

*2.3. High Throughput Sequencing of 16S rRNA Gene*

Samples were collected at the final stage of methane production for high-throughput sequencing of the 16S rRNA gene. DNA was extracted from each sample with E.Z.N.A.® soil DNA kit (Omega Bio-tek, Norcross, GA, USA), followed by polymerase chain reaction (PCR) amplification with primers 515F/806R. After purification and quantification, PCR products were sequenced by Illumina Miseq platform. Sequences were clustered by OTU based on a 97% similarity threshold and chimeras were eliminated. Based on the Silva 16S rRNA database (v138), the OTU representative sequences were annotated using RDP classifier, and the confidence threshold was set as 0.7 to obtain the taxonomic annotation results.

*2.4. Analytical Methods*

In order to quantitively compare the performance of each group, curve fitting was performed on the cumulative methane yield data over time using Originlab. The methane production in this study showed a two-stage pattern, and there was no obvious lag period in the first stage. Therefore, referring to the previous study [20], the first-order Model (1) and the modified Gompertz Model (2) were used to fit the methane production curve in the first and second stages of methane production, respectively. The overall Model (3) is obtained by combining the two models:

$$Y_t = Y_m(1 - exp(-kt)) \tag{1}$$

$$Y_t = Y_m exp\left\{-exp\left[\frac{\mu_m \cdot e}{Y_m}(\lambda - t) + 1\right]\right\} \tag{2}$$

$$Y_t = Y_{m1}(1 - exp(-kt)) + Y_{m2} exp\left\{-exp\left[\frac{\mu_m \cdot e}{Y_{m2}}(\lambda - t) + 1\right]\right\} \tag{3}$$

where $t$ = time from adding substrates in a batch (d); $Y_t$ = cumulated methane yield at $t$ (mL); $Y_m$ = specific methane yield (mL); $Y_{m1}$, $Y_{m2}$ = specific biogas yield at the first and

second stages (mL); $k$ = first-order digestion rate coefficient (1/d); $\lambda$ = length of lag phase (d); $\mu_m$ = maximum methane production rate (mL/d); and $e$ = Euler's number = 2.718.

TS and VS were measured according to standard methods for the examination of water and wastewater. Methane concentration was measured using gas chromatography with a thermal conductivity detector (Haixin, Shanghai, China). Gas chromatography with a flame ionization detector (Shimazu, Kyoto, Japan) and HP-FFAP column (Agilent, Santa Clara, CA, USA) was used to measure volatile fatty acids. The analysis of variance (ANOVA) was used to test the significance of results, and $p < 0.05$ was considered to be statistically significant.

## 3. Results

### 3.1. Effect of Hydrochar on Methane Production

The addition of hydrochar promoted methane production in all reactors with the three inocula, indicating that hydrochar can significantly enhance anaerobic digestion efficiency. Figure 1a shows the cumulative methane yield of the six reactor groups in the last batch as a function of time, and the parameters obtained by model fitting are shown in Table 2. The $R^2$ values of all fitting results are greater than 0.92, indicating that the model fits the methane production pattern well. For group G, F, and S, respectively, the lag period was shortened to 44.9%, 56.6%, and 16.1% of the control group when hydrochar was added. The maximum methane production rates $r_1$ and $r_2$ in the first and second stages were calculated by differentiating the fitted functions. The addition of hydrochar increased the maximum methane production rates in the second stage by 47.0%, 54.3%, and 53.6%. Hydrochar was effective in both reducing the lag period and increasing the methane production rates. However, the effect on each inoculum varied. Despite the consistent inoculum concentration of 4 g VS/L, there may be differences in the composition of the flora and microbial activity, thus leading to differences in the methane production patterns and promoting effects.

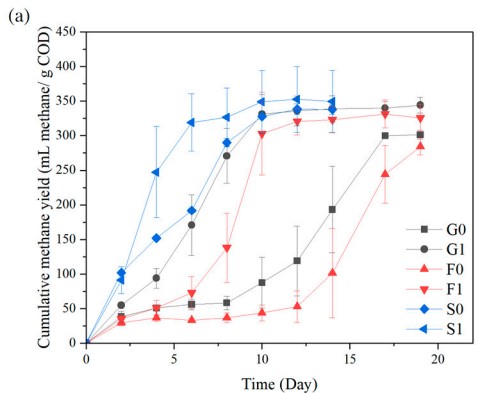
(a)

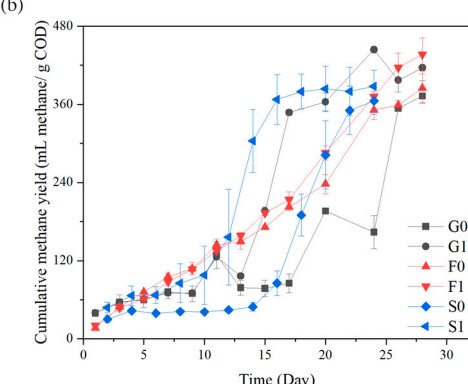
(b)

**Figure 1.** Cumulated methane yield. (**a**) Cumulative methane yield in the last batch; (**b**) Cumulative methane yield in the first batch.

**Table 2.** Reaction parameters obtained from model fitting.

| | $Y_{m1}$ mL Methane/g COD | $Y_{m2}$ mL Methane/g COD | $k$ 1/d | $\mu_m$ mL/d | $\lambda$ d | $Y_{m1} + Y_{m2}$ mL Methane/g | $R^2$ | $r_1$ mL/d | $r_2$ mL/d |
|---|---|---|---|---|---|---|---|---|---|
| G0 | 74.87 | 255.38 | 0.29 | 41.28 | 10.93 | 330.25 | 0.9574 | 20.52 | 41.74 |
| G1 | 129.42 | 214.54 | 0.30 | 55.60 | 4.90 | 343.96 | 0.9775 | 38.85 | 61.40 |
| F0 | 41.44 | 272.67 | 0.53 | 52.21 | 12.82 | 314.11 | 0.9484 | 21.87 | 52.21 |
| F1 | 93.84 | 237.03 | 0.23 | 77.54 | 7.25 | 330.87 | 0.9812 | 21.28 | 80.56 |
| S0 | 199.46 | 141.03 | 0.36 | 49.56 | 5.78 | 340.49 | 0.9989 | 71.31 | 55.76 |
| S1 | 0.00 | 347.99 | 0.00 | 85.69 | 0.93 | 347.99 | 0.9283 | / | 85.66 |

The methane production curves for F0, F1, and G0 with longer lag periods show that the reactor produced a small amount of methane in the early stage, followed by a lag

period in which the methane production increased moderately until the logarithmic growth period with rapid methane production. The methane produced in the early stage meant that the methanogens were active at this time and, considering that the microbes had been cultivated in the same conditions for three batches, the methanogens should have been activated. Then, with sufficient supply of glucose, the reactor should continue producing methane instead of going through a lag period of up to 13 days. Based on the data of VFA concentration (Figure 2), it was inferred that the methanogens were inhibited by the accumulation of VFAs, which was caused by the gap between the rates of VFA production and consumption. After a lag period, the methanogens slowly proliferated and initiated methanogenesis again. Gomes's study [21] addressed this two-phase methanogenesis pattern and its model fitting, regarding VFA accumulation as one of the reasons for diauxie.

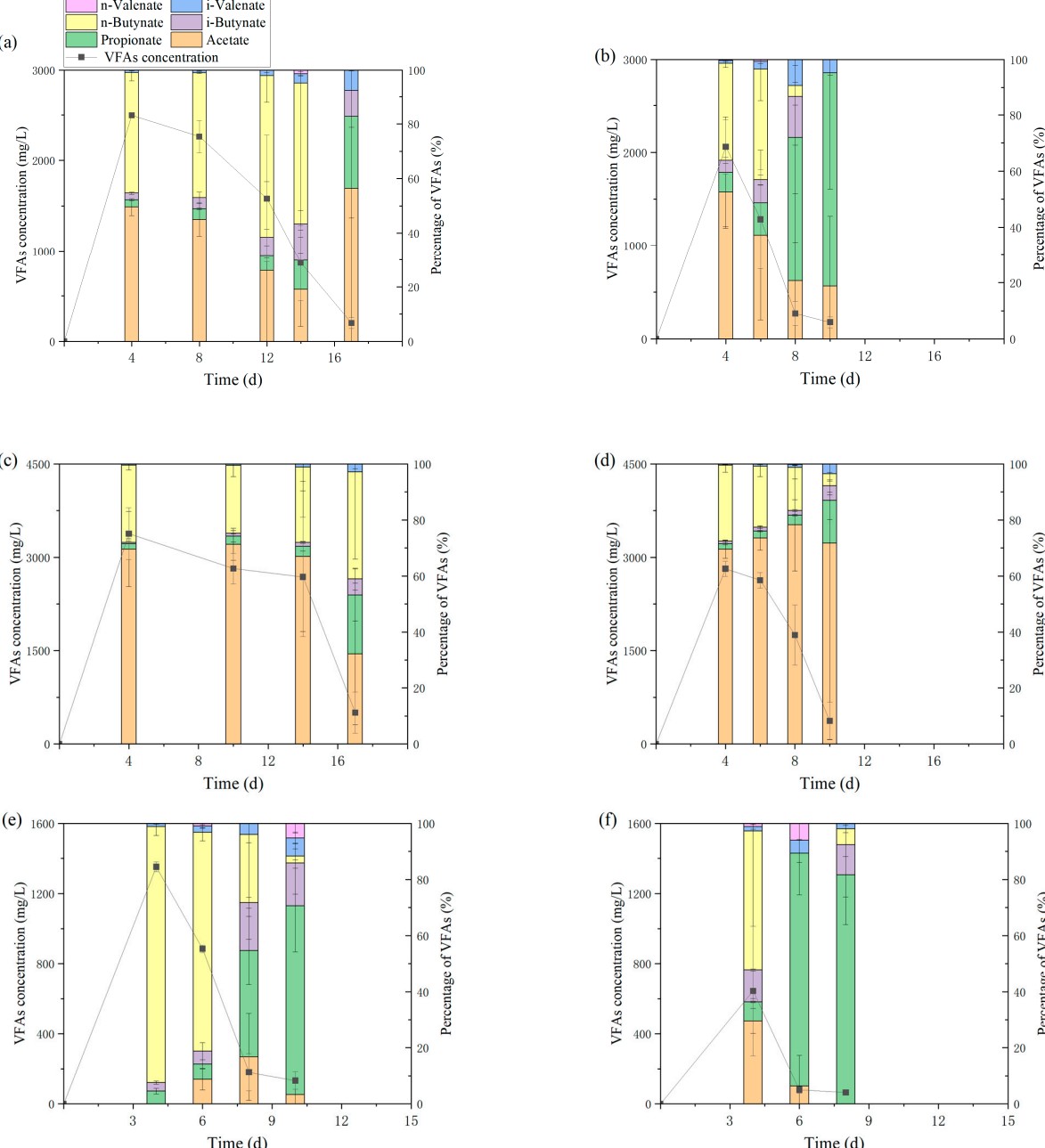

**Figure 2.** Concentration of VFAs in each group and the cumulative methane yield. (**a**–**f**): Group G0, G1, F0, F1, S0, S1.

The shortened lag period indicated that hydrochar alleviated the VFA inhibition, which was in accord with previous studies on biochar [22,23]. The fact that hydrochar did not significantly promote methane production of group F in the first batch (Figure 1b) supports the conclusion from the opposite side. During the first batch in F0, there was a linear growth in methane yield, and microbial sequencing revealed that the food waste digestate lacked glucose-degrading bacteria, and therefore the low rate of VFA production did not cause VFA inhibition. In subsequent batches, as the glucose-degrading bacteria proliferated, group F showed a similar methane production pattern as groups G and S, and then the addition of hydrochar showed a facilitative effect.

VFAs accumulated rapidly in the first 4 days, and the accumulated VFAs were dominated by acetate and n-butyrate. The F0 group with the longest lag period had the highest VFA and acetate concentrations of 3379 mg/L and 2353 mg/L, respectively, while the maximum n-butyrate concentration was found in S0 at 1234 mg/L. The maximum VFA concentration in the reactor containing hydrochar was lower than that in the control group, with the G, F, and S groups being 17.5%, 16.7%, and 52.4% lower than the corresponding control groups, respectively, which corroborated the results obtained from model fitting; G1 and F1 produced more methane in the first stage than G0 and F0, and the maximum methane production rate $r_1$ in the first stage of G1 was 89.5% higher than that of G0, indicating that the methanogens in the reactor with hydrochar had higher activity and produced more methane before VFAs accumulated. The situation was different in group S with sludge digestate as inoculum. Firstly, there was only one stage and no lag period in group S1 because of the higher microbial activity and different microbial composition of sludge digestate. Although there were two stages in the control group S0, the methane yield in the first stage was more than half of the total yield and the VFA concentration was 1352 mg/L, much lower than that in groups G and F. Secondly, the VFAs accumulated in the reactor of group S were dominated by butyrate solely. It is worth noting that no acetate was detected on day 4 in S0, while the concentration of butyrate decreased on day 6 as the concentration of acetate increased slightly, indicating that the rate of acetate production was lower than the rate of consumption during the first 4 days in S0. Methanogenesis slowed down due to the insufficient acetate supply until day 6 when the rate of butyrate decomposition increased. Therefore, the diauxie in S0 was not caused by the inhibition of VFAs, but by the blockage of the conversion of butyrate to acetate. In contrast, a small amount of acetate accumulated in S1 on day 4, and the concentration of butyrate was substantially lower than that in S0, suggesting that the addition of hydrochar increased the rate of butyrate degradation or reduced the production of butyrate. A similar phenomenon was found in the study by Wang et al. [3]. The acetate/butyrate ratio increased as the dope of biochar increased, and the conversion of butyrate to acetate occurred before, rather than simultaneously with, methane production. Wang speculated that this was due to the biochar assisting in the electron transfer of the process.

### 3.2. Effect of Hydrochar on Microbial Community

#### 3.2.1. Microbial Composition of the Inocula

The microbial composition of inocula from different sources varied and had distinct characteristics. Figure 3 shows the taxonomic classification of microorganisms in the inocula. The percentage of archaea was highest in the anaerobic granular sludge with 53.7%, followed by 7.1% in the sludge digestate and only 1.1% in the food waste digestate, which may explain the long lag period in group F. The dominant methanogens were *Methanosaeta*, *Methanobacterium*, and *Methanolinea*, and *Methanoculleus* was only present in the food waste digestate.

The relative abundance of *Bacteroidota* exceeded 10% in all three inocula, and the relative abundance of *Proteobacteria* ranged from 4.3% to 18.7%. *Bacteroidota* is common in the intestine and can utilize a wide range of substrate types and can break down polysaccharides such as cellulose [24]. The anaerobic species of *Proteobacteria* are mostly desulfurizing or sulfur-reducing bacteria, and also include the genus *Syntrophus*, which

can grow syntrophically with methanogens [25]. *Firmicutes* was the dominant phylum with 46.7% and 54.2% in granular sludge and food waste digestate, respectively, and less in sludge digestate. *Firmicutes* is related to the decomposition of many macromolecular organic substances [26]. *Chloroflexi* had a high proportion of 20.1% and 12.0%, respectively, in sludge digestate and granular sludge. *Chloroflexi* was lacking in the food waste digestate, but there was a higher abundance of *Cloacimonadota* (9.1%). *Acidobacteriota* was only abundant in the digested sludge (17.7%).

Analysis at the genus level revealed a large number of bacteria that were not classified or could not be identified to a genus, and analysis of the remaining identified genera revealed that the food waste digestate was rich in bacteria that break down proteins as well as participate in desulfurization; the sludge digestate was rich in bacteria that break down sugars and produce VFAs, and also contained some thermophilic anaerobic and aerobic bacteria, probably from aerobic reactors for wastewater treatment. Figure 4 is a heat map of the microbial composition of the three inocula, showing the top 50 bacteria and archaea in relative abundance. The dominant genera of sludge digestate and food waste digestate rarely overlap, indicating the distinct microbial composition of the inocula.

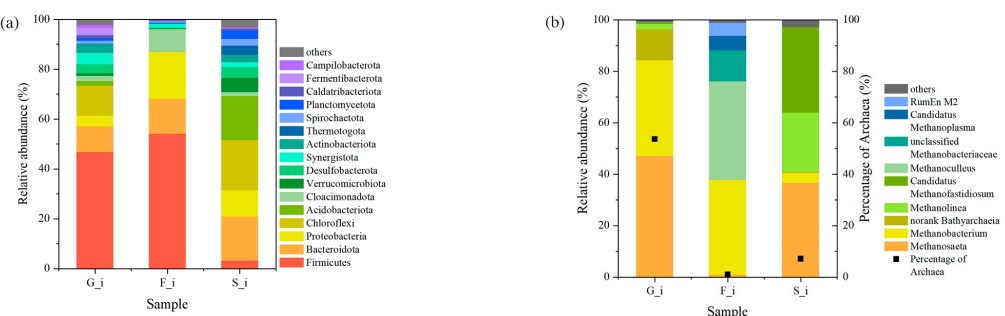

**Figure 3.** Taxonomic classification of microorganisms in the inocula. (**a**) Bacteria on phylum level; (**b**) archaea on genus level.

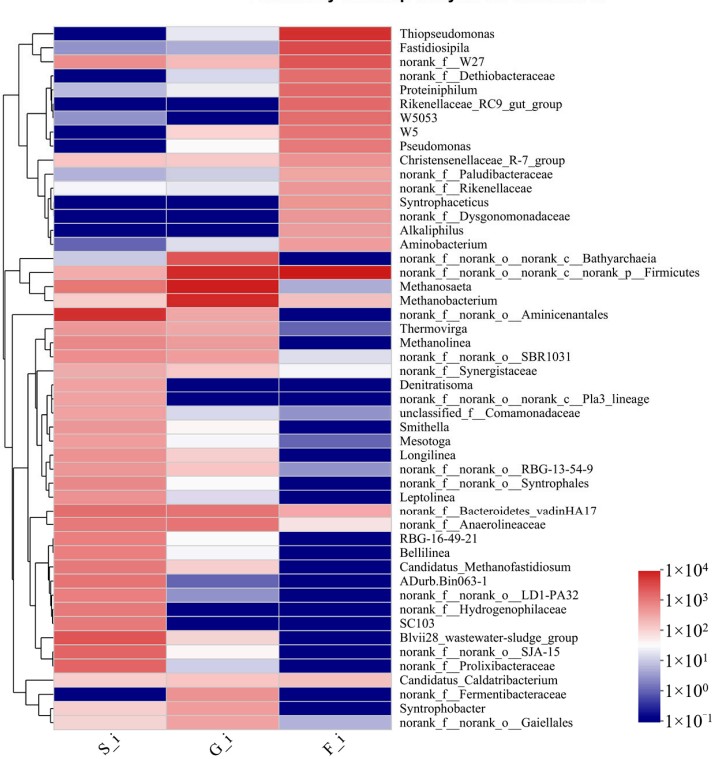

**Figure 4.** Heat map of the top 50 abundant microbes clustered by relative abundance (genus level).

### 3.2.2. Bacteria Composition at the End of Experiment

After cultivation with glucose as a substrate for four batches, *Firmicutes* dominated in all reactors (Figure 5). The inocula of groups G and F had a high proportion of *Firmicutes*, which was maintained after cultivation. The proportion of *Firmicutes* in the inoculum of group S was only 3.2%, which was increased to 38.5% and 49.2% after cultivation, indicating that, in this experiment, *Firmicutes* was the main phylum of bacteria that catabolized glucose and VFAs. *Bacteroidota* and *Chloroflexi*, which had high proportions in the inocula, did not change much in the proportions before and after cultivation. In addition, *Synergistoga*, which was present in small amounts in the inoculum, was enriched after cultivation, especially in the F group where the proportions reached 15.4% in F0 and 11.4% in F1.

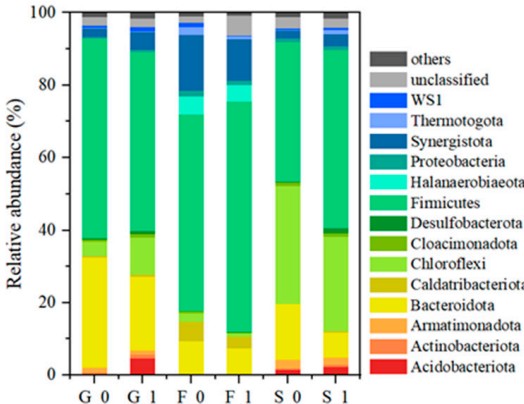

**Figure 5.** Taxanomic classification of bacteria on phylum level at the end of experiment.

Figure 6 shows the taxonomic classification and relative abundance of bacteria in each reactor at genus level. In this study, *Clostridium* was the major bacteria responsible for glucose degradation. *Clostridium sensu stricto* is the cluster of *Clostridium* most closely related to the type species *Clostridium butyricum* based on the 16S rRNA gene [27], and the suffix number represents its further classification by the Silva database. In this study, *Clostridium* refers to all *Clostridium sensu stricto*. *Clostridium* can utilize a wide range of substrate types, with typical metabolites being acetate and butyrate, sometimes producing other VFAs and hydrogen. Some species can consume $CO_2$ and hydrogen for the isotype production of acetate [28]. *Clostridium* was rare in the inocula (less than 0.01% in G and F and less than 0.1% in S), but it dominated all reactors at the end of the experiments, with relative abundances of G0 27.8%, G1 32.4%, F0 32.6%, F1 29.0%, S0 24.9%, and S1 32.6%. Hydrochar increased the relative abundance of *Clostridium* in the G and S groups but slightly decreased it in group F. The composition of *Clostridium* varied from reactor to reactor, and the composition of other bacteria also varied.

In group G, *Clostridium sensu stricto 5* was the most abundant on genus level (G0 23.7%, G1 17.7%). Those reactors with hydrochar had a higher proportion of *Clostrdium sensu stricto 1* and *3*. Several genera of *Bacteroidota* with low contents in the inoculum were enriched after cultivation, mainly *Proteiniphilum* and *Lentimicrobium*, and they were not identified on species level. Their relatives, *P. saccharofermentans* and *L. saccharophilum*, catabolize glucose, with acetate as the main product and a small yield of propionate and hydrogen [29,30]. Compared with the control group, the bacteria enriched by the addition of hydrochar were *Aminicenantales* (4.7% vs. 0.3%), *SBR1031* (2.5% vs. 0.3%), and some genera of *Anaerolineaceae*.

In group F, the relative abundance of *Clostridium sensu stricto 1* was the highest (F0 14.6%, F1 15.1%). *Clostridium sensu stricto 3* was enriched in reactors with hydrochar (F1 7.4%, F0 1.0%). *Aminobacterium* and *Acetomicrobium* (family *Synergistaceae*) were enriched, and both were more abundant in the control group. *Aminobacterium* and *Acetomicrobium* catabolize amino acids and glucose, respectively, to produce acetate and hydrogen, and prefer syntrophic growth with hydrogen-consuming methanogens [31,32]. The no

rank *Halobacteroidaceae* and *Syntrophomonadaceae* were also enriched, and the relative abundance was similar in the two groups. The bacteria enriched by adding hydrochar was *Lachnospiraceae* (F1 2.9% vs. F0 0.6%), which is taxonomically close to *Clostridium* and was one of the main intestinal flora, producing mainly acetate and other VFAs [33]. In addition, the F1 group with the addition of hydrochar had more bacteria that could not be classified in the existing databases, such as 6.8% *Firmicutes* and 5.5% bacteria whose phylum could not be classified.

In group S, *Clostridium sensu stricto 1* and *5* were dominant with relative abundances of 16.6% in S0 and 11.3% in S1, and 6.8% in S0 and 18.3% in S1, respectively. *Longilinea* was significantly enriched, with relative abundance of S0 27.2% and S1 17.9%. *Longilinea arvoryzae*, the type species, could not utilize glucose, VFAs, or pyruvate [34]. *Longilinea* enriched in this experiment has not been isolated before. According to the experimental conditions, it should be capable of degrading glucose or VFAs. The addition of hydrochar increased the relative abundance of *SBR1031* (S1 2.9% vs. S0 0.7%) and no rank *Syntrophomonadaceae* (S1 4.1% vs. S0 0.6%).

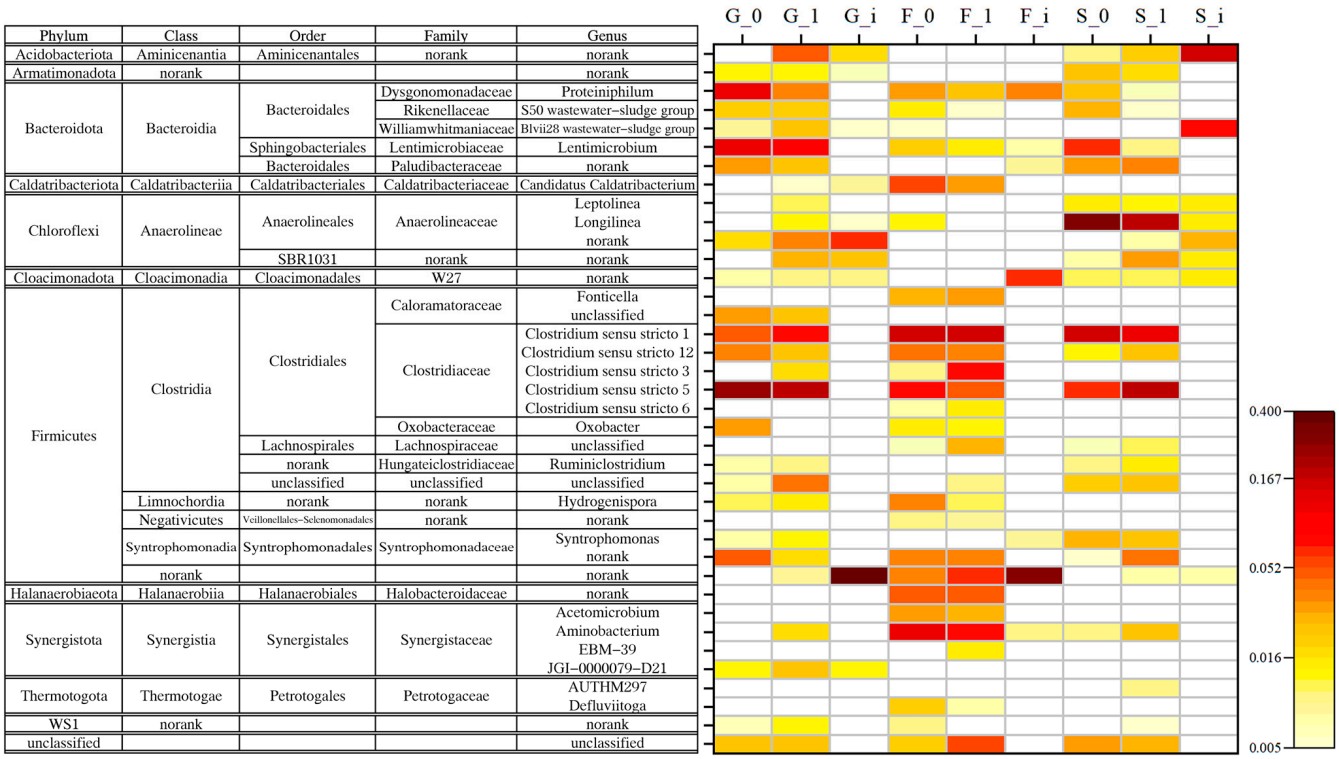

**Figure 6.** Taxanomic classification of bacteria on genus level at the end of experiment.

### 3.2.3. Effect of the Addition of Hydrochar on Bacteria Composition

The enhancement of DIET is the primary consideration when analyzing the reasons why hydrochar promotes anaerobic digestion. Due to the advantages of DIET in reaction rate compared with MIET, the bacteria participating in DIET should proliferate more. Therefore, the following investigation focuses on the bacteria whose abundance increased after the addition of hydrochar in each group.

*Aminicenantales* was present in a small amount in groups G and S. It is a syntrophic bacterium that produces acetate and hydrogen by consuming carbohydrates and proteins. Genetic analysis suggests that it may possess type IV pili [35], which are essential for microbes to transfer electrons extracellularly, which can support the speculation that it gained advantage through DIET. Some genera of *Anaerolineaceae* were present in groups G and S. Previous studies have suggested that *Anaerolineaceae* may be involved in extracellular electron transfer based on much evidence, including: the possession and transcription of the

pilA gene that encodes pilus assembly proteins, the ability to transfer electrons to ferric iron, and its enrichment on the electrodes of microbial fuel cells. *Clostridium* is dominant in all reactors. Some *Clostridium*, such as *C. thermobutyricum* belonging to *Clostridium sensu stricto 7*, and *C. acetobutylicum* belonging to *Clostridium sensu stricto 11*, can reduce Fe(III) [28,36], which means they have the potential to participate in DIET. However, in this study, only *Clostridium sensu stricto 3* increased in reactors with hydrochar, and the effect of hydrochar on the other groups is uncertain. In this study, the three groups enriched the same no rank *Syntrophomonadaceae* (OTU1248), which is in the same family as *Syntrophomonas*, and several studies have suggested that the latter can participate in DIET [4,37]. It was less than 0.01% in all three inocula, and its enrichment means that it played a role in glucose metabolism or VFA metabolism. However, the effects of the addition of hydrochar on its relative abundance in the three groups were inconsistent, and thus, whether it participated in DIET remains unclear. On the other hand, *Syntrophomonas* was present in small amounts but was not enriched in this experiment. As for *SBR1031* in groups G and S and *Lachnospiraceae* in group F, there is no existing evidence of their DIET potential. However, considering the ability of carbon materials transferring electrons between cells, which may realize DIET between microbes that are incapable of spontaneously conducting DIET, the possibility of *SBR1031* and *Lachnospiraceae* participating in DIET cannot be excluded.

Most of the bacteria enriched in the reactor at the end of the experiment were rare in the inocula, except for *Aminicenantales* and some genera of *Anaerolineae* in groups G and S, as well as *Proteiniphilum* and no rank *Firmicutes* in group F. Among them, four groups were stimulated by hydrochar, namely *Aminicenantales*, no rank *Anaerolineaceae*, *SBR1031*, and no rank *Firmicutes*. Apart from these four groups, the only bacterium that was consistently enriched by hydrochar was *Clostrdium sensu strict 3*. The other bacteria were either below 1% or were inversely affected by hydrochar in different groups. It can be concluded that there was a considerable overlap between bacteria that were highly abundant in the inoculum and maintained high relative abundance after the experiment, and bacteria that were increased by the addition of hydrochar. This may be because direct contact is necessary for hydrochar to promote microbial metabolism, and microorganisms that were abundant in the inoculum had a greater chance of attaching to the surface of hydrochar, while the later enriched microorganisms had a smaller chance of reaching the hydrochar as the surface of the hydrochar had been occupied, and thus were less affected by the hydrochar. The adherence and attachment ability of *SBR1031* and *Anaerolineaceae* can also explain their enrichment with the addition of hydrochar [38], as hydrochar provided more surface available for attachment.

In addition, more no rank and unclassified bacteria at the genus level were present in the reactor with hydrochar. Among these bacteria that have not been isolated and investigated, there may be species that can be promoted by hydrochar and participate in DIET. In order to identify these species and analyze their metabolic functions, metagenome and meta-transcriptome approaches should be applied in future studies.

Most of the dominant bacterial genera in this study had a higher abundance in the control group, indicating that the addition of hydrochar did not promote the proliferation of these dominant bacteria, which also means that the reactor with hydrochar had a higher evenness. Table 3 shows the Simpson index and Shannon index of each reactor. The higher the Simpson index, the lower the evenness of the species, while the Shannon index is the opposite. It can be seen that the reactors with hydrochar had higher evenness regardless of the inoculum. The higher evenness of the flora helped the anaerobic reactor to cope with environmental fluctuations [39]. The Chao index represents the estimated OTU/species/genera number of the community. A higher Chao index indicates a larger community richness.

**Table 3.** Alpha diversity indexes of the bacteria community in each group.

| Sample | G0 | G1 | F0 | F1 | S0 | S1 |
|---|---|---|---|---|---|---|
| Simpson Index | 0.0985 | 0.0638 | 0.0680 | 0.0607 | 0.1262 | 0.0922 |
| Shannon Index | 3.0802 | 3.4853 | 3.3378 | 3.4276 | 3.0309 | 3.2626 |
| Chao Index | 241.85 | 253.49 | 239.57 | 222.30 | 365.60 | 376.82 |

### 3.2.4. Archaea Composition at the End of the Experiment

At the end of the experiment, the proportion of archaea ranged from 13.9% to 36.7% (Figure 7). The proportion of archaea in granular sludge was 53.7%, which decreased after cultivation, while the proportions in group F and S increased after cultivation. The proportion of archaea increased with the addition of hydrochar by 9.2, 3.1, and 2.0 percentage points in groups G, F, and S.

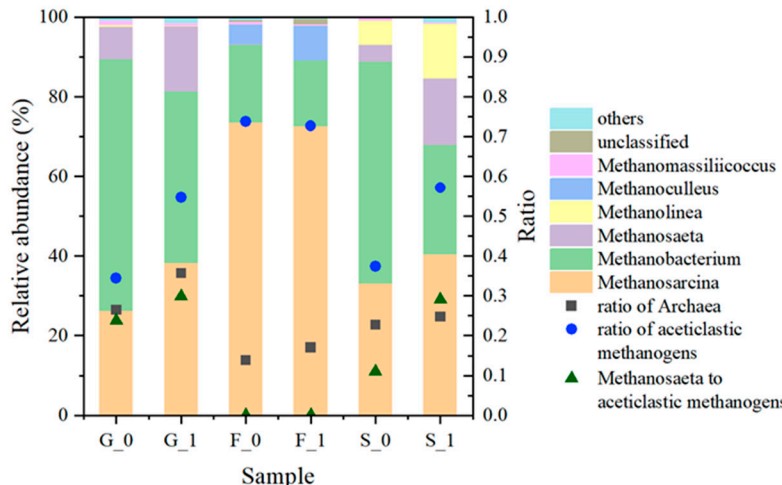

**Figure 7.** Taxanomic classification of archaea on genus level at the end of experiment.

The major methanogens in all groups were *Methanosarcina* and *Methanobacterium*. *Methanosaeta* was not present in the F group, but accounted for about 16% of group G1 and S1, more than the corresponding control groups, indicating that hydrochar promoted the growth of *Methanosaeta*. *Methanosaeta harundinaceais* showed direct electron transport ability under pure co-culture conditions [14]. Several studies have found its enrichment with the addition of biochar, suggesting that biochar promoted DIET [40,41]. However, it was noteworthy that *Methanosaeta* accounted for 47.3% and 36.7% of the archaea in the inocula of groups G and S, respectively, while it did not maintain a dominant position. In contrast, *Methanosarcina* was rare in all three inocula (<1%), but predominated in all groups at the end of the experiment. In a number of studies using glucose or acetate as substrates, *Methanosarcina* appeared to be lacking in the inoculum, but it gradually became dominant as the reaction proceeded [23,42,43]. *Methanosarcina* and *Methanosaeta*, both being acetoclastic methanogens, have a competitive relationship. The study of Conklin et al. found that *Methanosaeta* was prone to dominate in steady low concentrations of acetate, while when the concentration of acetate was high due to intermittent feeding, *Methanosarcina*, with a high half-saturation coefficient, is more likely to dominate [44], which is in line with the conditions and phenomena of this experiment. It is speculated that the enrichment of *Methanosarcina* is due to the high acetate concentration caused by sequential batch cultivation. After all, hydrochar promoted the growth of both *Methanosarcina* and *Methanosaeta*.

### 3.2.5. Influence of the Difference in Inocula on Microbial Community

PCA analysis (Figure 8) was performed on the microbial composition (including bacteria and archaea) of the inocula and samples after cultivation, and the distance between the dots in the figure reflects the extent of the difference between the samples. It can be seen that the microbial composition of the three inocula converged after cultivation, due to the four sequential batches that enriched all reactors with bacteria using glucose, butyrate, and acetate as the main carbon sources. The difference between the hydrochar and control groups was not significant, but there was a significant difference between the three inoculum groups(R = 0.828, P = 0.001). Although hydrochar promoted methane production in all three groups, it did not significantly affect the overall microbial composition.

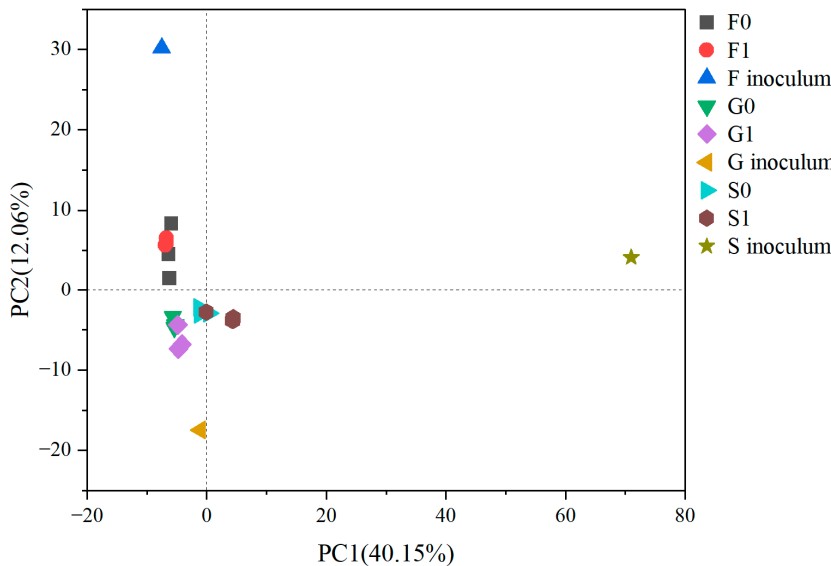

**Figure 8.** PCA analysis on the microbial composition (genus level).

The difference in inoculum had limited effect on the final microbial composition. Firstly, the inoculum determined the presence or absence of certain key microorganisms. For example, *Anaerolineae* and *Methanosaeta* were abundant in group G and S, indicating that the experimental conditions favored their growth, but they were not present in group F due to their absence in the food waste digestate. Which microorganisms dominated in the inoculum had little effect on the final microbial composition, as any microorganisms that were not adapted to the experimental conditions would rapidly decrease regardless of their abundance in the inoculum, such as no rank *Firmicutes* and *Thiopseudomonas* (not shown in Figure 6). In contrast, some microorganisms with very low abundance in the inoculum may eventually dominate, such as *Clostridium*, *Syntrophomonadaceae*, and *Methanosarcina*. Although the dominant microorganisms were the same in all three groups, bacteria with higher relative abundance but without absolute dominance varied depending on the inoculum.

## 4. Discussion

The three different inocula with various microbial composition all lacked *Trichococcus* and *Azospira*, which were enriched in earlier studies under similar conditions. Some bacteria enriched by the addition of conductive materials in earlier studies, such as *Geobacter* and *Thermvirga*, were not enriched either, indicating that hydrochar may affect a wide range of microorganisms other than those mentioned above. In addition, in this experiment, hydrochar did not cause significant changes in the overall composition of microorganisms, and bacteria increased by the addition of hydrochar only accounted for a low proportion (around 10%) of the total. Therefore, the mechanism by which hydrochar promoted anaerobic digestion in this study is worth discussing.

Hydrochar increased the proportion of methanogens, especially the proportion of acetoclastic methanogens *Methanosarcina* and *Methanosaeta*, that possessed the ability of DIET [14,45]. It has been demonstrated that accumulation of VFAs led to a lag period, which was significantly shortened by the addition of hydrochar, indicating that hydrochar may increase methanogenic activity at high VFA concentrations through DIET, which could explain the promoting effect of hydrochar. However, this explanation is insufficient. Firstly, the acetoclastic methanogens in group F did not increase with the addition of hydrochar. Secondly, DIET requires the participation of both electron-donating bacteria and electron-accepting methanogens, and the insignificant changes in bacteria composition is suspicious. In this regard, two conjectures were put forward: (1) the bacteria involved in DIET were few, but played an important role; (2) hydrochar had other effects besides enhancing DIET.

There have been other studies where the bacteria community did not change significantly with the addition of materials. Peng et al., for example, used magnetite and activated carbon to enhance the methane yield from waste sludge by 20%. Peng believed that magnetite accelerated the destruction of cell walls by promoting the growth of three groups of iron-reducing and sulfur-reducing bacteria, which altogether accounted for less than 10% [46]. Zhao et al. increased methane yield by more than 30% by adding biochar and pointed out that microorganisms attached to the carbon surface accounted for less than 5% of the total [5]. A number of studies have found that the composition of microorganisms attached to the surface of carbon materials was quite different from that of suspended microorganisms, and more functional microorganisms were attached to the surface of carbon materials [6,16,47]. Therefore, it can be speculated that the hydrochar promoted the metabolism of a small number of the surrounding microorganisms, which remained active under acid inhibition and consumed VFAs, accelerating the recovery from acid inhibition.

Fe(III) can act as the electron acceptor to consume the excess NAD(P)H produced during glycolysis, thereby reducing the production of butyrate and hydrogen [36]. Biochar also has a certain ability to accept electrons, and the addition of hydrochar in group S avoided the accumulation of butyrate. This may be due to the fact that hydrochar played a role similar to ferric iron as an electron intermediate involved in VFA metabolism and methanogenesis, rather than assisting DIET between cells through conductivity. In fact, the electrical conductivity of biochar is very weak, and it has been proposed that the ability of biochar and hydrochar to promote anaerobic digestion is correlated with the abundance of oxygen-containing functional groups on the surface and the electron accepting and donating capacity [2,48]. The phenomenon of group S in this study confirmed this theory from a certain perspective.

The adsorption capacity of hydrochar can also promote methanogenesis, but it is vital only when the substrate contains toxic substances [49]. The function of hydrochar as a support for microbial attachment is also of interest because the hydrochar applied in this study was powdered and easy to aggregate in water, which can help microbials to aggregate and resist acid inhibition.

There are some limitations to this study. Similar to many previous studies, most of the microorganisms in the reactor could not be identified on a species level and, as a result, their functions could only be inferred based on the species in the same genus or even the same family or order. However, even different species within the same genus have different metabolic pathways. The present study has limitations in microbial function identification, and the emerging metagenomic and other omics tools have great advantages in this regard. Studies on the isolation and investigation of more anaerobic microorganisms are also urgently needed.

## 5. Conclusions

Hydrochar enhanced methanogenesis in anaerobic digestion without the enrichment of several key bacteria involved in DIET, as reported by previous studies. The facilitation effect of hydrochar varied among reactors with different inocula. For granular sludge and acclimated food waste digestate, the addition of hydrochar alleviated the inhibitory effect



of high VFA concentration on methanogenesis, shortened the lag period of methanogenesis by about half, and increased the methanogenesis rate. For fresh food waste digestate, VFA accumulation did not occur, and hydrochar had no significant promoting effect. For sludge digestate, hydrochar promoted the conversion of butyrate to acetate and prevented the accumulation of butyrate while increasing the methanogenesis rate. After 86 days and 4 cycles of sequential batch cultivation, the microbial compositions of reactors with different inocula converged. In all reactors, *Clostridium* was the dominant genus that degraded glucose to produce VFAs, and *Methanosarcina* and *Methanobacterium* were the major methanogens. The microbial composition of the inoculum had limited effect on the final microbial composition. The inoculum determined the presence or absence of key microorganisms, and microorganisms with a higher proportion in the inoculum were more likely to adhere to the hydrochar and maintain a relatively high abundance at the end of experiment. The addition of hydrochar had limited changes in bacterial composition, and hydrochar only increased the abundance of bacteria with lower content. The addition of hydrochar increased the proportion of acetoclastic methanogens in archaea, and especially promoted the growth of *Methanosaeta*. It was speculated that hydrochar acted as an electron intermediate and supported microbial aggregation, while the possibility that hydrochar promoted DIET cannot be ruled out.

**Supplementary Materials:** The following supporting information can be downloaded at: https://www.mdpi.com/article/10.3390/fermentation9050433/s1, Figure S1: Reactor configuration and experimental processes; Figure S2: Mass balance of carbon; Figure S3: Model fitting curves; Table S1: The characteristics of the hydrochar; Table S2: Correction of methane yield.

**Author Contributions:** Conceptualization, G.L. and J.S.; methodology, J.S.; formal analysis, J.S.; investigation, J.S.; resources, S.Z. and G.L.; writing—original draft preparation, J.S.; writing—review and editing, G.L.; visualization, J.S.; supervision, G.L. and S.Z.; funding acquisition, G.L. All authors have read and agreed to the published version of the manuscript.

**Funding:** This work was financially supported by the National Natural Science Foundation of China (31970117), and the Science and Technology Commission of Shanghai Municipality (22ZR1405900).

**Institutional Review Board Statement:** Not applicable.

**Informed Consent Statement:** Not applicable.

**Data Availability Statement:** Data sharing is not applicable to this article.

**Conflicts of Interest:** The authors declare no conflict of interest. The funders had no role in the design of the study; in the collection, analyses, or interpretation of data; in the writing of the manuscript; or in the decision to publish the results.

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
