# Peer review of "The Role of Hydrochar in Promoting Methane Production from Anaerobic Digestion with Different Inocula"

_fermentation, doi:10.3390/fermentation9050433_

Round 1

Reviewer 1 Report

Method

Reactor setup:

How was the gas collected? was this an automated system?

Why a large headspace?

Line 142: What GC column was used?

Figure 1: By saying first batch and last batch, are they done at different times?

What tell you it was over the first 2 days? I would say it was longer than that. its also confirmed in your model.

Line 170: What do you mean by methane production paused? remember after lag is log phase. paused means no gas production. was this seen in your daily analysis? if so please state.

Line 172: The idea of the presence of the methanogen and lag phase should be based on the experiments with hydrochar where no lag phase was observed.

Line177: Do methanogen consume VFA? you need to state that the accumulation of VFAs inhibit methanogens. 

Line 180: Reference Gomes.

Line 206: Also due to the substrate composition compared to the other 2 digestate.

Line 228: The concentrates of methanogens in the various digestate is a clear indication why S (sludge) had basically no lag phase and why F (food) had the lowest. therefore, one can predict the stages the inoculum was at.

At what stages of the digesters were the G,S and F digestate collected. this could also explain for the long lag phases, and high VFAs in some reactors.  

By looking at the high VFAs can you say this is also due to the fact that acetogenesis is rate limiting and the process is accelerated by the hydrochar. remember VFA are converted to acetic acid, Hydrogen and carbon dioxide, which are then converted to methane by methanogens. Only when the conditions are right do they start their work.

The quality of the English is excellent but with very minor effort. 

Author Response

Response to Reviewer 1 Comments

Point 1: How was the gas collected? was this an automated system? Why a large headspace?

Response 1: The serum bottles used as reactors were sealed with rubber caps, so the biogas produced during AD was contained in the bottles. Automated systems were not applied, and the composition of the headspace gas was detected manually every 2~3 days. The large headspace was set to avoid high gas pressure. The gas in the reactors was released after detection in order to maintain a relatively low pressure. 

Point 2: Line 142: What GC column was used?

Response 2: The GC column was HP-FFAP.

Point 3: Figure 1: By saying first batch and last batch, are they done at different times?

Response 3: Yes, they were done at different times. 4 batches were done in sequence. A figure explaining the reactors configuration was supplied in the SM.

Point 4: What tell you it was over the first 2 days? I would say it was longer than that. its also confirmed in your model.

Response 4: The description was not proper, and it is now revised as “in the early stage”.

Point 5: Line 170: What do you mean by methane production paused? remember after lag is log phase. paused means no gas production. was this seen in your daily analysis? if so please state.

Response 5: “pause” was not the proper word and is revised to “increase moderately”. Only in F0 did methane production paused, which could be proved by data. 

Point 6: Line 172: The idea of the presence of the methanogen and lag phase should be based on the experiments with hydrochar where no lag phase was observed.

Response 6: Experiments with more delicately controlled conditions is more convincing, however, we believe the inferences and conclusions of this paper are reasonable and acceptable.

Point 7: Line177: Do methanogen consume VFA? you need to state that the accumulation of VFAs inhibit methanogens.

Response 7: No, methanogen do not consume VFAs other than acetate and formate, so we fixed the expression. It was stated in the manuscript that accumulation of VFAs inhibited methanogens but maybe not very clearly. So, this sentence is now revised as “Based on the data of VFAs concentration (Figure 2), it was inferred that the methanogens were inhibited by the accumulation of VFAs, which was caused by the gap between the rates of VFAs production and consumption.”

Point 8: Line 180: Reference Gomes.

Response 8: OK, the reference number is moved forward.

Point 9: Line 206: Also due to the substrate composition compared to the other 2 digestate.

Response 9: Yes, the difference in microbial composition (which was due to different substrates) is now mentioned as one of the reasons.

Point 10: Line 228: The concentrates of methanogens in the various digestate is a clear indication why S (sludge) had basically no lag phase and why F (food) had the lowest. therefore, one can predict the stages the inoculum was at.

Response 10: Yes, this conclusion is added.

Point 11: At what stages of the digesters were the G,S and F digestate collected. this could also explain for the long lag phases, and high VFAs in some reactors. 

Response 11: If you are asking about the inocula, we are sorry that we don’t know from which part of the reactors were the inocula collected but we are quite sure that they were all from one-stage reactors. Based on the 16s rRNA sequencing results, we suppose the food wastes digestate was collected near the feed port. Granular sludge of group G was from a UASB reactor and there should be minor difference in microbial compositions among different locations in the reactor.

Point 12: By looking at the high VFAs can you say this is also due to the fact that acetogenesis is rate limiting and the process is accelerated by the hydrochar. remember VFA are converted to acetic acid, Hydrogen and carbon dioxide, which are then converted to methane by methanogens. Only when the conditions are right do they start their work.

Response 12: Yes, that is the case in group S0.

Reviewer 2 Report

In this study hydrochar was used to promote methane production from anaerobic digestion. In this study inocula was introduced from three distinct sources to anaerobic sequential batch reactors and the microbial community was analyzed by 16S rRNA gene sequencing. Results of the study may have important application in the design and operation of anaerobic digestion for wastewater treatment.  Authors may wish to consider the following comments in revision of their manuscript.

1.      Lab scale reactor was used in the study.  Please comment on the scale up factor for full scale application.

2.      Please comment on the limitations of the study.

3.      Please include some cost for production of hydrochar.

4.      Please perform mass balance of carbon for the conducted study.

5.      Please perform mass balance on solids for the conducted study.

6.      Please plot TSS (mg/L) vs time of the batch reactor run.

7.      Synthetic wastewater using glucose was used in the batch study. Please comment whether results may be applicable to real wastewaters which may contain organic and inorganic pollutants.

8.      Please report important factors which need to be considered in the design of the treatment system based on results obtained from conducted study.

English is acceptable.

Reviewer 3 Report

The authors set the stage to investigate the effect of hydrochar on methane production from anaerobic digestion and the differences in enriched bacteria. To investigate the effect of inoculum on the microbial composition and compare the promotion effects of hydrochar on different microorganisms, inocula from three different sources are used. Hydrochar showed enhanced methane productivity in all three inocula, by increasing microbial activity at high acid concentrations. Authors speculated that hydrochar act as electron intermediate and supported microbial aggregation. This study seems interesting, but there are some issues that have to be addressed     1) The effect of hydrochar in cumulated methane yield in the last batch (Figure 1) seems to be significant for G0-G1 and F0-F1 pairs. However, it has a minor effect on S0-S1 pair. In addition the methane yield is higher in S0-S1 pair (compared to the other 2 pairs).  How the authors explain this phenomenon?   2) Chao index (Table 2) is not analysed. What this index represents?  

Author Response

Response to Reviewer 4 Comments

 Point 1: The effect of hydrochar in cumulated methane yield in the last batch (Figure 1) seems to be significant for G0-G1 and F0-F1 pairs. However, it has a minor effect on S0-S1 pair. In addition the methane yield is higher in S0-S1 pair (compared to the other 2 pairs).  How the authors explain this phenomenon?  

Response 1: As for the different effect on S group, hydrochar could alleviate inhibition of VFA, but the concentration of VFAs were relatively low in group S. As the result, the promotive effect of hydrochar in S group was less significant. As for the different methane yields, the substrate loss caused by sampling could partially explain this phenomenon. The sampling frequency was 2-3 days, and 1 mL liquid was taken out each time. The total liquid volume was 50 mL, and 6 mL of S and 8 mL of G and F was extracted. For reactors that reacted slower and finished later, more liquid was extracted and there was a higher content of glucose and VFAs in the extracted liquid. We tried to fix this error by estimating the amount of substrate that was extracted, and the result was presented in supplementary material. After the correction, the difference in methane yield among groups was reduced to about the half, but not totally eliminated. Another possible reason is that less carbon was converted to cellular contents when there was hydrochar.

Point 2: Chao index (Table 2) is not analysed. What this index represents?  

Response 2: Chao index represents the estimated OTU/species/genera number of the community. A higher Chao index indicates a bigger community richness.

Reviewer 4 Report

The manuscript "The role of hydrochar in promoting methane production from anaerobic digestion with different inocula" is a basic study on anaerobic digestion, which evaluated the effect of hydrochar addition. I believe that the paper can be accepted after major revisions.

1) The hydrochar should be characterized. Some results of the characteristics should be presented in the paper.

2) The initial composition of the sludges used should be presented in a Table.

3) There is a lack in the physicochemical characterization of the reactors.

4) I could not understand the reactors configuration. In Figure 1 the authors used the term "last batch" and "first batch". This should be better clarified. Maybe a figure in the methodology with the reators confiuguration can be an alternative to better clarify the process.

5) Figure 1 presents the methane yield, but the results were expressed in mL. Methane yield should be expressed as mL methane per g TVS (added or consumed). Please, revise in the whole manscript.

6) Include a figure with the fitted curve for the methane production in the kinetic models adopted. This can be added as a SM.

7) The unit of the parameters preseted in Table 1 should be included.

8) Figure 2 is poor, and the data should be presented in a better way. Please, improve the figure.

9) In the section 3.2.5, authors should include the Pearsons correlation coefficients between all variables, and better thiscuss the correlation between methane, VFA, and microbial community.

10) In the whole text, include a space between the word and the number of the reference. Example: hydrochar[1] should be hydrochar [1].  

Beside these points, the paper is in the scope of Fermentation. Make sure to upload the figures in a good resolution and with the same font used in the manuscript.

Author Response

Response to Reviewer 4 Comments

Point 1: The hydrochar should be characterized. Some results of the characteristics should be presented in the paper.

Response 1: Characteristics of hydrochar are supplied in the SM.

Point 2: The initial composition of the sludges used should be presented in a Table.

Response 2: A table has been added.

Point 3: There is a lack in the physicochemical characterization of the reactors.

Response 3: pH was not monitored during experiment, however, NaHCO3 was added as pH buffer. According to our previous study under similar condition, the pH of the reactors were supposed to maintain over 6.5.

Point 4: I could not understand the reactors configuration. In Figure 1 the authors used the term "last batch" and "first batch". This should be better clarified. Maybe a figure in the methodology with the reators confiuguration can be an alternative to better clarify the process.

Response 4: Some of the descriptions about reactor configuration have been modified, and a figure has been added to the SM.

Point 5: Figure 1 presents the methane yield, but the results were expressed in mL. Methane yield should be expressed as mL methane per g TVS (added or consumed). Please, revise in the whole manscript.

Response 5: The unit is now revised, but as mL CH4 / g COD instead of TVS, because the substrate was glucose and COD is more commonly used for glucose.

Point 6: Include a figure with the fitted curve for the methane production in the kinetic models adopted. This can be added as a SM.

Response 6: The fitted curves have been added to the SM.

Point 7: The unit of the parameters preseted in Table 1 should be included.

Response 7: The units are now included and the fitted parameters have changed due to the change in the unit of methane yield.

Point 8: Figure 2 is poor, and the data should be presented in a better way. Please, improve the figure.

Response 8: The figure is modified.

Point 9: In the section 3.2.5, authors should include the Pearsons correlation coefficients between all variables, and better thiscuss the correlation between methane, VFA, and microbial community.

Response 9: The only two variables which were set different among the groups were: the presence of hydrochar and the compositions of the inocula, either of which is numerical. Other variables (initial pH, substrate composition, concentration of the inoculum, temperature) were set the same. So, Pearsons correlation analysis is not viable.

Point 10:  In the whole text, include a space between the word and the number of the reference. Example: hydrochar[1] should be hydrochar [1].  

Response 10: The formation is fixed. Thank you for your correction and advice.

Round 2

Reviewer 3 Report

-

Reviewer 4 Report

The paper can be accepted.